# Design and Fabrication of a Novel Poly-Si Microhotplate with Heat Compensation Structure

**DOI:** 10.3390/mi13122090

**Published:** 2022-11-27

**Authors:** Xiaorui Lu, Jiahui Liu, Guowei Han, Chaowei Si, Yongmei Zhao, Zhongxuan Hou, Yongkang Zhang, Jin Ning, Fuhua Yang

**Affiliations:** 1Engineering Research Center for Semiconductor Integrated Technology, Institute of Semiconductors, Chinese Academy of Sciences, Beijing 100083, China; 2Center of Materials Science and Optoelectronics Engineering, University of Chinese Academy of Sciences, Beijing 100049, China; 3School of Electronic, Electrical and Communication Engineering, University of Chinese Academy of Sciences, Beijing 100049, China; 4State Key Laboratory of Transducer Technology, Chinese Academy of Sciences, Beijing 100083, China

**Keywords:** microhotplates, poly-Si, LPCVD, uniform temperature distribution, power consumption, response time, infrared image

## Abstract

I Microhotplates are critical devices in various MEMS sensors that could provide appropriate operating temperatures. In this paper, a novel design of poly-Si membrane microhotplates with a heat compensation structure was reported. The main objective of this work was to design and fabricate the poly-Si microhotplate, and the thermal and electrical performance of the microhotplates were also investigated. The poly-Si resistive heater was deposited by LPCVD, and phosphorous doping was applied by in situ doping process to reduce the resistance of poly-Si. In order to obtain a uniform temperature distribution, a series of S-shaped compensation structures were fabricated at the edge of the resistive heater. LPCVD SiNx layers deposited on both sides of poly-Si were used as both the mechanical supporting layer and the electrical isolation layer. The Pt electrode was fabricated on the top of the microhotplate for temperature detection. The area of the heating membrane was 1 mm × 1 mm. Various parameters of the different size devices were simulated and measured, including temperature distribution, power consumption, thermal expansion and response time. The simulation and electrical–thermal measurement results were reported. For microhotplates with a heat compensation structure, the membrane temperature reached 811.7 °C when the applied voltage was 5.5 V at a heating power of 148.3 mW. A 3.8 V DC voltage was applied to measure the temperature distribution; the maximum temperature was 397.6 °C, and the area where the temperature reached 90% covered about 73.8% when the applied voltage was 3.8 V at a heating power of 70.8 mW. The heating response time was 17 ms while the microhotplate was heated to 400 °C from room temperature, and the cooling response time was 32 ms while the device was recovered to room temperature. This microhotplate has many advantages, such as uniform temperature distribution, low power consumption and fast response, which are suitable for MEMS gas sensors, humidity sensors, gas flow sensors, etc.

## 1. Introduction

In recent years, MEMS microhotplates have attracted much attention due to their numerous advantages, such as low power consumption, rapid response, high level of integration and compatibility with CMOS technology [1,2]. Microhotplates were widely used as the key component of MEMS sensors, for instance, gas sensors [3,4,5,6], humidity sensors [7], gas flow sensors [8], etc. Gas sensors typically operate at high temperatures in a range of tens to hundreds of degrees Celsius (generally below 400 °C) [9,10,11,12,13,14,15,16,17,18,19,20,21,22,23,24,25] and require uniform heating to active the sensitive material, which places great demands on the electro–thermal properties of microhotplates to improve gas sensitivity [9]. A comparison of various microhotplates was summarized in Table 1.

For gas sensors, the resistive heaters and detection structure generally used the same set of Pt electrodes [17,18,19,20,21,22,23,24,25]. On the one hand, the Pt electrode was used as a heater. On the other hand, a Pt electrode was applied to acquire the response signal by detecting current and resistance changes. Because the reaction of the target gas would lead to the temperature change, and the resistance of the Pt electrode varied in proportion to these temperature changes. However, to obtain a higher heating temperature, lower resistance of heaters was desirable, while the resistance of detection electrodes should be as large as possible for an accurate response. Therefore, it was essential to enhance the sensitivity by separating the resistive heater and detection electrodes.

Generally, the SiN*_x_* membrane was deposited as the electrical isolation layer [22,23,24,25,26]. The common deposition process of SiN*_x_* included PECVD (plasma-enhanced chemical vapor deposition) and LPCVD (low-pressure chemical vapor deposition). Usually, Pt was chosen to fabricate the heater resistance because of its excellent linear relationship between the resistance and temperature, which was important for the gas sensor. Therefore, PECVD SiN*_x_* films were widely used in microhotplates because of their lower process temperature below 400 °C [23,26], whereas the process temperature of LPCVD was usually above 800 °C, which would lead to the damage of Pt resistive heater because of its poor adhesion and high surface tension, such as peeling and voids [27,28]. Under the high temperature and voltage conditions, the PECVD SiN*_x_* membrane would generate a higher leak current of several microamperes, which was of the same order of magnitude as the response signal [29], leading to the invalidation of gas sensors. In addition, the thermal conductivity of PECVD SiN*_x_* would be modified irreversibly caused by Si-H and N-H bonds breaking down and all hydrogen diffusing, which limited its usage at the higher temperature [30]. Compared with the PECVD SiN*_x_* membrane, the LPCVD SiN*_x_* membrane had lots of advantages, such as being more compact, having fewer defects and having better electrical insulation performance [31,32]. 

In order to avoid the Pt electrode damage caused by peeling, sintering and grain coarsening, many methods were presented. Various metal/metal oxide layers were deposited between the Pt layer and substrate, such as Ti [27,28,33], TiO_2_ [27,29], Zr [17,18,29,33], Ta [17,18,33], Al_2_O_3_ [28,29,34] and Hf [18], which were used as adhesion and barrier layer. However, this method could lead to the degradation of the bimetal system due to metal interdiffusion and chemical interaction between them [35]. Another approach to increase the thermal stability of Pt thin film was achieved by doping the refractory oxide materials (ZrO_2_ [36,37,38], HfO_2_ [37,39], CeO_2_ [39]) or alloying additives (Zr [38,39], Rh [35,38,39], Ir [40]). The dopants could stabilize the grain boundaries and decrease the diffusion mobility of Pt. However, the dopants would introduce nonlinear variation in the temperature-resistance relationship, which limited the appliance in gas detection.

Poly-Si had good thermal stability and excellent adhesion, which could endure the high temperature in the LPCVD process and annealing. Therefore, it was practicable to use poly-Si as resistive heaters instead of Pt so that the LPCVD SiN*_x_* membrane could be used for better electrical isolation and preventing the deterioration of the sensor [32]. In addition, heating uniformity was also an important parameter for the gas sensor, which had a significant influence on the sensitivity. Different microheater geometry investigated so far by different researchers [41,42,43,44]. The S-shaped compensation structure was an effective measure to improve the temperature at the edge of the membrane. The compensation structure could balance the thermal loss at the edge of the membrane caused by heat conduction, convection and radiation. With the introduction of the compensation structure, the thermal uniformity would improve several times [45,46,47,48,49,50]. However, the structure dimension and conductivity of poly-Si could be further optimized to improve the heating performance of microhotplates.

In this paper, a low-resistance poly-Si microhotplate with a temperature-compensated structure was designed and fabricated. This device was designed to operate at 400 °C stably and constantly. In situ doping was used to improve the conductivity of poly-Si. At the edge of the resistive heater, a series of S-shaped compensation structures were fabricated to improve the thermal uniformity of the membrane. The electrical and thermal measurements were investigated to evaluate the heating characteristics of microhotplates. This study also demonstrated the response performance of the microhotplates when operating at 400 °C.

**Table 1 micromachines-13-02090-t001:** Comparison of various microhotplates.

Structure	Membrane Size/Heater Size (μm^2^)	Maximum Temperature (°C)	Power Consumption (mW)	Ref.
SiN/Ti/Pt/SiN/Al_2_O_3_	100 × 100	350	20	[4]
SiN/Ti/Pt/SiO_2_	50 × 50	350	6	[7]
Si/SiN/Pt/SiN	500 × 500	400	70	[9]
Pt/PECVD-SiN/LPCVD-SiN/Si	300 × 300	400	39	[26]
${\mathrm{SiO}}_{2}/\mathrm{SiN}/{\mathrm{SiO}}_{2}/TiO_{2-\delta }$ $/\mathrm{Pt}/TiO_{2-\delta }$	500 × 500	800	120	[27]
SiN/Cr/CrN/Pt/CrN/Cr/ SiO/SiN/Glass/SiN	2500 × 2500	498	2350	[45]
Insulation nitride/heater/ SiN/SiO/Si	120 × 120	421	30	[46]
Pt/SiO/TaN/SiO/SiN/SiO/ Si/SiO/SiN/SiO	300 × 300	450	100	[47]
SiO_2_/ITO/SiO_2_/Si	290 × 290	350	386	[48]
SiN*_x_*/poly-Si/SiN*_x_*	1000 × 1000	811.7	148.3	This work

## 2. Materials and Methods

### 2.1. Fabrication Details

The microhotplates were fabricated on the 380 μm double-side polished *p*-type <100> Si wafer. At first, a 500 nm LPCVD SiN*_x_* membrane was deposited on the Si substrate, which was used as a mechanical supporting layer, wet etching mask and the etching stop layer for the wet etching of the silicon substrate. Then a 300 nm LPCVD thick low resistance poly-Si layer was deposited, as shown in Figure 1a. Phosphorous doping was applied by in situ doping process to reduce the resistance of poly-Si. After the deposition, the poly-Si layer was patterned to fabricate the resistive heater by SF_6_ ICP (inductively coupled plasma) etching, as shown in Figure 1b. The mixture of buffered oxide etchant (BOE) and nitric acid was applied to remove the poly-Si on the backside of the wafer. Then, the wafer was annealed at 950 °C for half an hour in a nitrogen and oxygen atmosphere for the recrystallization of poly-Si. Subsequently, the same SiN*_x_* membrane was deposited on the top of the poly-Si layer, which was used for the electrical isolation between the resistive heater and the follow-up Pt detection electrodes. These two SiN*_x_* layers were vertically symmetric to keep the membrane flat, as shown in Figure 1c. 

As shown in Figure 1d, the contacting holes in the SiN*_x_* layer were patterned in by photolithography and CHF_3_ plasma etching process. Whereafter, platinum (Pt) temperature sensors and contacting pads of resistive heaters were fabricated by photolithography, magnetron sputtering of 50 nm Ti/300 nm Pt, and lifting-off. Furthermore, to reduce the contact resistance, 50 nm Ti/300 nm Au was sputtered to fabricate the thickened electrodes at the pads. The thickened electrodes were also fabricated by the lifting-off process.

Finally, because LPCVD was a kind of double-side deposition process, the SiN*_x_* layer on the backside of the silicon wafer was etched by CHF_3_ plasma to fabricate the wet etching window. Residual SiN*_x_* worked as the hard mask in wetting etching. An 80 °C 20% KOH solution was used to perform anisotropic wetting etching of the silicon substrate with continuous stirring so that the platform would be suspended completely after 6.5 h to etch about 380 μm. All the above process flows are shown in Figure 1. The microhotplates with the conventional planar structure were called Sample−1, and the microhotplates with compensation structure were called Sample−2, in which the critical dimension (d) was valued at 20 μm. The scanning electron microscopies (SEM) of both samples are shown in Figure 2.

### 2.2. Design

MEMS metal oxide gas sensors normally need to be provided with an operating temperature to activate charge transfer between sensitive materials and the absorbed gas molecules, which ranges from 150 °C to 400 °C. Furthermore, the temperature distribution should be as uniform as possible so that the sensitive materials can be evenly heated to obtain a consistent response.

#### 2.2.1. Poly-Si Resistive Heaters

To obtain a more uniform temperature distribution, we designed a series of heat consumption structures at the edge of resistive heaters. Generally, for the film microhotplates, the temperature decreased rapidly from the center to the edge of the membrane due to the higher thermal conductivity of the silicon substrate. The thermal conductivity of silicon was about 147 W/m*K, which was much larger than that of SiN*_x_* (usually thought to be less than 10 W/m*K). Therefore, the heat generated from poly-Si resistive heaters is transmitted to the silicon substrate and dissipated, leading to a significant decrease in temperature around the outer areas of the heater. To compensate for such heat loss, it was a feasible solution to increase the heat consumption at the edge of the membrane. As a result, the additional heat generated prevented the dramatic decline in temperature and contributed to a more uniform temperature distribution.

Figure 3 shows the schematic diagram of the poly-Si resistive heaters with designed power compensation structures (in the green box). As shown, the resistive heaters were made of 10 identical resistances connected in parallel. Each resistance had fold shape compensation at both ends. For a single resistance, these structures and the major part (in the black box) were connected in series so that the currents along this resistance are equal. According to Joule’s law:(1)$$P=I^{2}\times R$$

Since the current was constant, the power consumption was proportional to resistance. The compensation structures increased the additional resistance at the edge of poly-Si heaters, which introduced a series of smaller and denser S-shaped pattern designs. The extra heat generated by the increase in localization resistance could make up for the heat loss at the boundary between the membrane and the silicon substrate, resulting in a higher temperature in the outer area than the conventional planar heaters, driving a more uniform temperature distribution.

#### 2.2.2. Pt Temperature Sensor

The Pt resistance was fabricated on the top of the membrane to measure the temperature of microhotplates. Pt resistance change was linearly related to the temperature variation, so the Pt resistance worked as a temperature sensor.

The Ti layers were used to enhance the adhesion of Pt or Au to the substrate and form an effective ohmic metallization at the pads of the Pt sensor and poly-Si heaters.

The area of the microhotplate, heater membrane and Pt temperature sensor were 2.5 mm × 2.5 mm, 1 mm × 1 mm and 600 μm × 600 μm, respectively.

## 3. Simulation and Experiments

### 3.1. Simulation

The finite element simulation with the electric–thermal–solid coupled field was applied to estimate the surface temperature and stress–strain distribution of microhotplates by COMSOL Multiphysics 5.5. The structure of the microhotplate was modeled, and the Pt temperature sensor was simplified as a rectangular block. The temperature distribution, thermal expansion and expansion stress of planar structure and compensation structure microhotplates were simulated to obtain an optimized structure critical dimension (d) for the following fabrication. Additionally, all other dimensions of the module stayed the same with the fabricated pattern mentioned above; the exploded and vertical views are shown in Figure 4. The material parameters used in the simulation are shown in Table 2; all the parameters were applied to the thin film material instead of that of the bulk. By using the four probes method to test the square resistance, the conductivity of poly silicon was $1.6\times 10^{3}\;\Omega \cdot \mathrm{cm}$. In addition, thermal convection had a lot of contributions to heat transfer. In the simulation, the external nature convection of the upside of the horizontal plate was applied. The air convection coefficient and the environment temperature were set as 6 W·m^−2^·K^−1^ [51] and 25 °C, respectively. Radiation heat transfer was also significant at these temperatures, and surface-to-ambient radiation was applied. The emissivity of the poly-Si was 0.6, and the emissivity of the Pt at several hundreds of degrees was 0.1 [52]; the radiation constant was defined as the Stefan–Boltzmann constant 5.67 × 10^8^ W·m^−2^·K^−4^.

Figure 5 shows the simulation results of temperature distribution with different structures while the applied voltage was 3.8 V. Figure 5a shows the thermal distribution of planar resistive heater; Figure 5b–d shows the distribution of resistive heater with compensation structures; and d values are 10, 20 and 30 μm, respectively. The area where the temperature reached 90% of the maximum temperature ($T_{90}$ area) was used to evaluate the thermal uniformity, which was indicated by the red color.

For the planar convention heater shown in Figure 5a, the simulation result indicated that the high-temperature area was highly concentrated in the center of the microheater with a maximum temperature of 427 °C, while the $T_{90}$ area was a rounded square of around 470 μm, covering about 23.2% of the membrane.

When the value of *d* is 10 μm, as shown in Figure 5b, the highest temperature on the platform reached 426 °C. $T_{90}$ area was concentrated at a region of an approximate ellipse, in which the horizontal and vertical axes were about 550 μm and 580 μm, respectively, covering about 32.2% of the heater. The power consumption was 123.9 mW.

Figure 5c,d show the thermal distribution while d valued for 20 and 30 μm, respectively. Both heaters had a uniformly $T_{90}$ area in a range of about 950 × 700 μm^2^, which took up more than 73.8% of the membrane. The $T_{90}$ area has improved more than three times in comparison to the conventional planar structures. Moreover, the maximum temperature was 421 °C whiles d=20 μm, 44 °C higher than when the value was 30 μm. On the contrary, the power consumptions while d valued for 20 μm and 30 μm were 70.8 mW and 67.3 mW, respectively. The light decline of the highest temperature and power consumption was due to the increase in the total resistance of resistive heaters.

Considering the thermal distribution, maximum temperature and power consumption, the compensation structure with the figure dimension value at 20 μm had a better electrothermal performance.

In addition, Figure 6 shows the comparison of the thermal expansion of microhotplates with conventional structures and novel structures. As shown, compared with the planar structure microhotplate, the thermal expansion and expansion stress of the center of the microhotplate with compensation structure were decreased to 0.12 μm and 64 MPa from 0.17 μm and 153 MPa, respectively. Additionally, the maximum stress also decreased to 207 MPa from 501 MPa. It was because the fabricated structure could release the stress caused by thermal expansion, which efficiently improved the mechanical stability of microhotplates.

### 3.2. Experiments

The electric performance was measured by using the probe station and B1500A semiconductor analyzer. By heating to 250 °C from RT and testing the resistance, the TCR of Pt temperature sensor was measured first, and the conversion relation between temperature and Pt resistance was obtained in a wider temperature range. Then, DC voltage was applied to the resistive heater via a probe station, and the operating temperature was calculated through converting. 

In the thermal testing, the microhotplates were glued on the PCB, and gold wires were bonded to connect chips and PCB. A DC voltage source was connected to the PCB for charging. The applied voltage was sweeping between 0 and 6 V. Thermal image and video were used to analyze the temperature distribution and heating/cooling performance, respectively.

All the test system diagrams mentioned above are shown in Figure 7. The glowing sample bonded with PCB is also shown in Figure 7.

A scanning electron microscope (SEM, Nova NanoSEM650, FEI Company, Hillsborough, OR, USA) was used to observe the microstructure morphology. The temperature coefficient of resistance (TCR) of the Pt temperature sensor was measured using the high-temperature semiconductor measurement system, which consisted of the probe station (PW800s, ADV Technology Co., Pasadena, CA, USA), B1500A Semiconductor Device Analyzer (Keysight Technologies, Inc., Santa Rosa, CA, USA) and electric furnace. This system could raise the temperature of the samples and measure their resistance. The heating and electric performance of microhotplates were measured by using an infrared imaging device (FOTRIC 280, Fotric Co., Shanghai, CN) and B1500A for analyzing the temperature distribution and power consumption. A dc voltage current source/monitor was used for the electricity supply and long-term dependability evaluation.

## 4. Results and Discussion

After 24 h aging by applying a voltage of 4 V and several times switching to obtain a stable heating performance, the infrared thermograms of Sample−1 and Sample−2 were captured to evaluate the heating uniformity and shown in Figure 8. Driven by a DC voltage of 3.8 V, the maximum temperature in the thermogram of Sample−2 was 396.2 °C, which was nearly identical to the result of the Pt sensor. The maximum temperature of Sample−1 was slightly decreased compared with the value measured by the Pt sensor. This difference was related to the focus position of a thermal infrared camera.

Figure 8 shows the significant difference in the thermal distribution between the two samples. The $T_{90}$ area was the bright yellow area. For the Sample−1, the $T_{90}$ area was a rounded square concentrated on half of the area of the whole membrane, and the temperature difference between the edge and center of the membrane was nearly 200 °C. In contrast, the $T_{90}$ area of Sample−2 covered almost the entire membrane, and the temperature difference was less than 50 °C. The $T_{90}$ area of the heater with compensation structure was almost three times larger than that of the conventional planar heater. The comparison between the test and simulation results is likewise shown in Figure 8, where it can be seen that the experimental data agreed with the simulated results at a high level.

The relationship between heating temperature and applied voltage of the microhotplate with compensation structure was also measured. As shown in Figure 9a, the TCR of Pt, obtained by linear fitting, had a value of 3770 ppm/K. It can be seen the Pt temperature sensor had a high linear correlation between temperature and resistance, which could be applied to accurately calculate the actual heating temperature of the poly-Si heater. Figure 9b shows the voltage-temperature characteristics of Sample−2 and the resistance of the Pt sensor. The membrane temperature was 397.6 °C when the applied voltage was 3.8 V, and power consumption was 73.4 mW. Additionally, the membrane temperature reached 811.7 °C when the applied voltage was 5.5 V, and power consumption was 148.3 mW. No hysteresis was observed during several voltage sweeps. With the further increase in voltage to 6 V and the further rise of temperature, the membrane broke after stopping the heating, and the temperature reached nearly 1000 °C, as projected by the resistance value of Pt.

Figure 10 shows response performance by applying a step voltage. The heating response time was defined as the time at which the microhotplate was heated to 90% of the maximum temperature from room temperature. Correspondingly, the cooling response time was defined as the time at which the microhotplate was cooled to 10% of the maximum temperature from the maximum temperature. Figure 10a,b shows that when the membrane temperature heated up to 400 °C from room temperature, the heating response time was 17 ms, and the cooling response time from 400 °C to room temperature was 32 ms. In general, as the heating temperature increased, the heating response time became shorter while the cooling response time became longer, and temperature changed faster when heating from temperature to cooling toward room temperature, as shown in Figure 10c. At the beginning of heating, the resistance of poly-Si was relatively low, which resulted in a high current and a large heating power. After several milliseconds, the current and power decreased to a constant value with the temperature increasing. When the voltage increased, the heating power at the beginning became higher because of the peak current. Therefore, the heating response time became shorter. Conversely, during the cooling process, when the membrane temperature decreased, the heat flux decreased simultaneously. With the initial temperature increasing, the total heat increased, leading to a longer cooling response time.

As shown in Figure 10d, the cooling time had no significant change while the micro-hotplate was operating at 400 °C for an hour, which verified that the heater did not affect the temperature of the area around the membrane. The simulation results and infrared thermal images also indicated that the chip temperature was slightly above room temperature. Therefore, gas sensors using these microhotplates have no need for special packaging techniques to avoid high temperatures [53].

## 5. Conclusions

In this study, a novel poly-Si membrane micro-hotplate was designed and fabricated via a MEMS technological process. The detection structure and resistive heater were separated to solve their contradiction. LPCVD SiN*_x_* membrane was used for better electrical isolation. In order to withstand the process temperature of LPCVD, the resistive heater was fabricated with low-resistance poly-Si. The folder shape compensation structure at the film edge had higher resistance and generated more heat than the major part. This additional heat balanced the heat loss caused by the strong convection from the edge of the membrane to the Si substrate. Therefore, the poly-Si resistive heater with the temperature compensation structure could improve the heating uniformity effectively. The compensation structure with the critical dimension of 20 μm was the most optimized design with a maximum temperature of 397.6 °C and the $T_{90}$ area coverage of 73.8% when the applied voltage was 3.8 V and the heating power was 73.4 mW. When the applied voltage was 5.5 V, the membrane temperature reached 811.7 °C, and power consumption was 148.3 mW. The heating response time was 17 ms while the microhotplate was heated to 400 °C from room temperature, and the cooling response time was 32 ms while the device was recovered to room temperature. A long operating time had no signification influence on the temperature except for the membrane. Due to all the properties mentioned, the microhotplates with compensation structure were suitable for thin film gas sensors.

## Figures and Tables

**Figure 1 micromachines-13-02090-f001:**
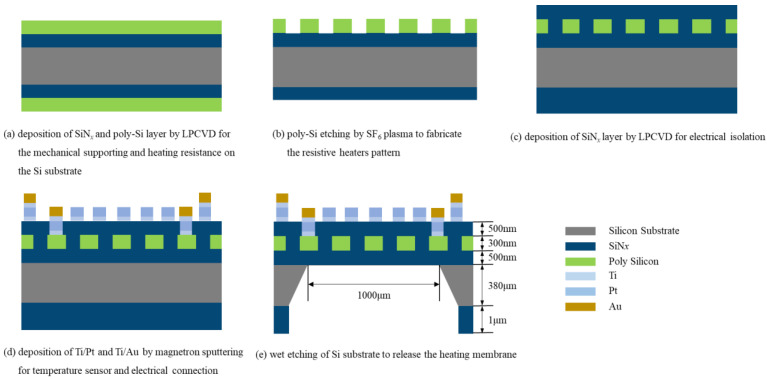
Poly-Si microhotplate fabrication process flow.

**Figure 2 micromachines-13-02090-f002:**
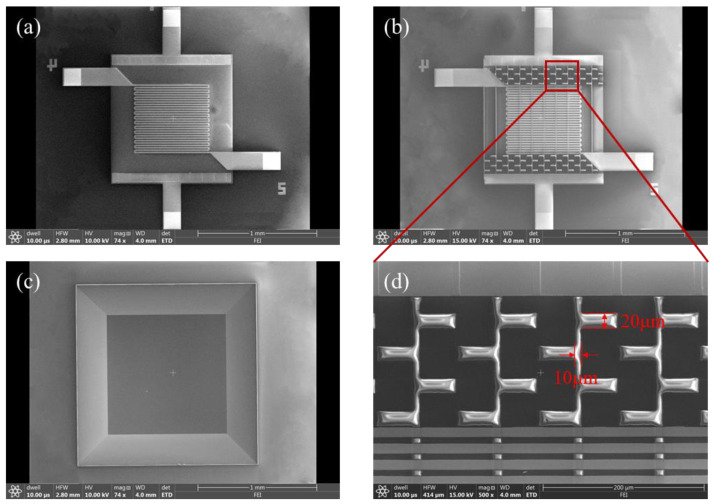
SEM of microhotplates: (**a**) top view of Sample−1, (**b**) top view of Sample−2, (**c**) cavity formed by wetting etching at the back side and (**d**) detailed view of compensation structure of Sample−2.

**Figure 3 micromachines-13-02090-f003:**
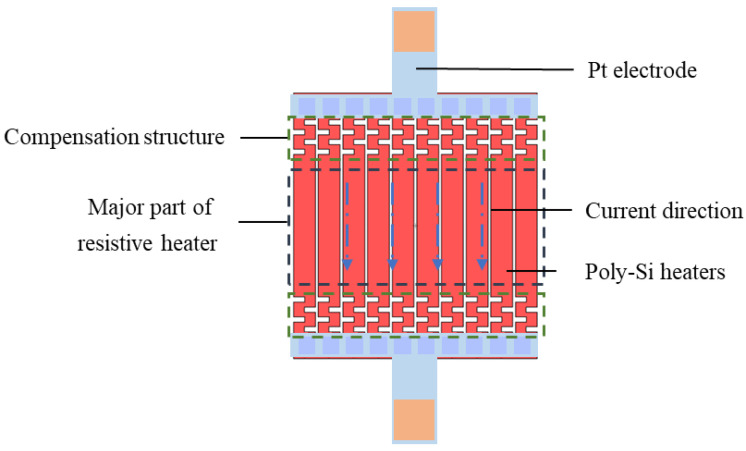
Schematic diagram of the poly-Si resistive heaters with designed heat compensation structures.

**Figure 4 micromachines-13-02090-f004:**
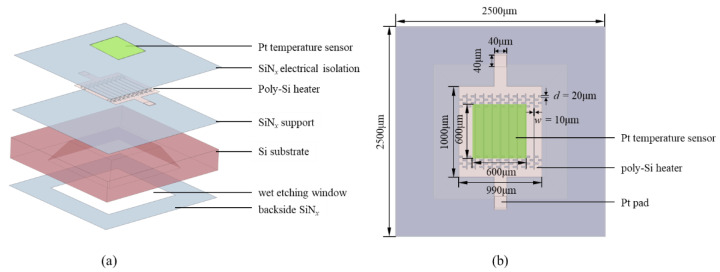
Simulation module: (**a**) exploded view, (**b**) vertical view.

**Figure 5 micromachines-13-02090-f005:**
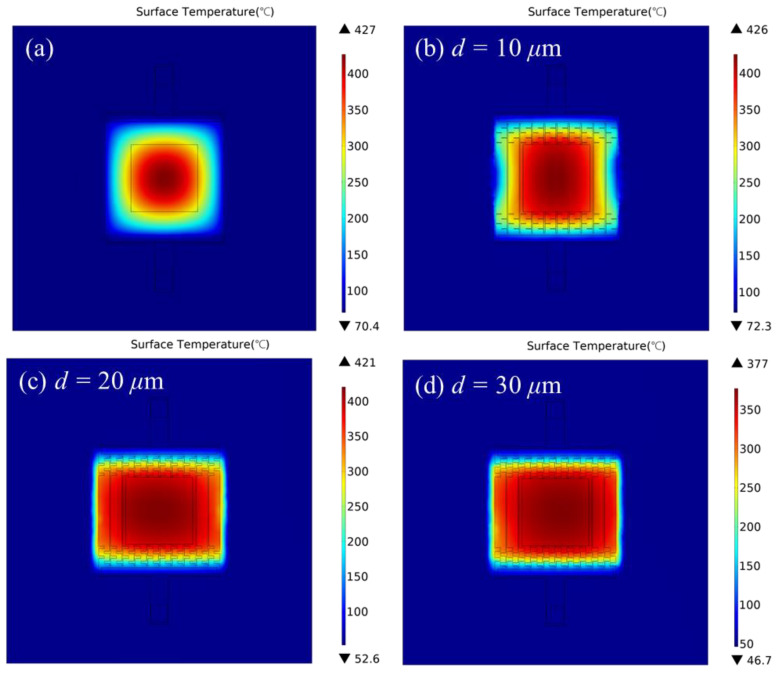
Simulation results of surface temperature and thermal distribution.

**Figure 6 micromachines-13-02090-f006:**
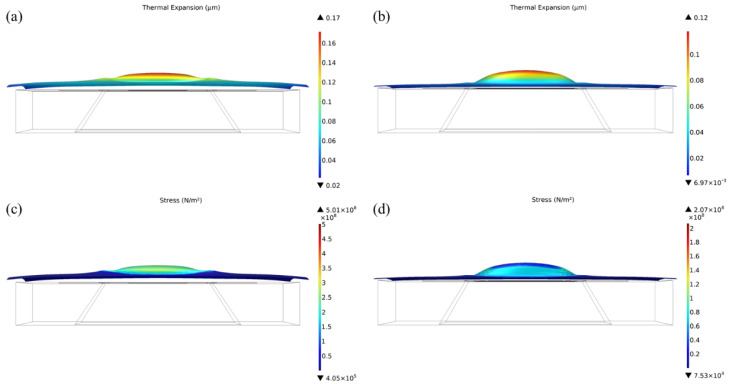
Thermal expansion and stress while heating to 400 °C: (**a**,**c**) microhotplate with conventional planar structure; (**b**,**d**) microhotplate with a 20 μm compensation structure.

**Figure 7 micromachines-13-02090-f007:**
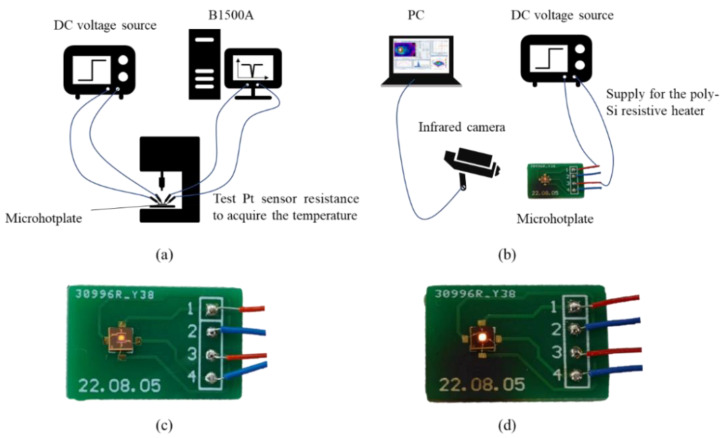
Test system connection: (**a**) electric performance; (**b**) thermal performance; (**c**) Sample−1 and (**d**) Sample−2 bonded with PCB.

**Figure 8 micromachines-13-02090-f008:**
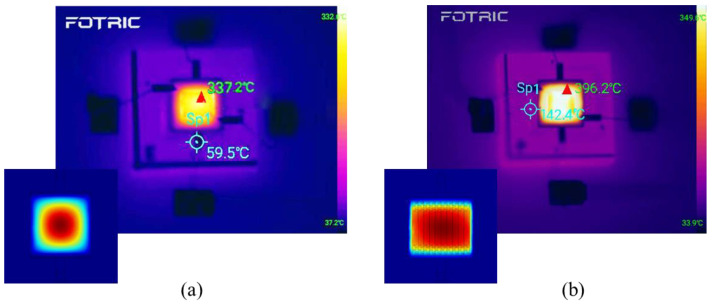
Temperature distribution measured by $T_{90}$ area in the infrared thermogram: (**a**) conventional planar poly-Si microhotplate; (**b**) poly-Si microhotplate with compensation structure.

**Figure 9 micromachines-13-02090-f009:**
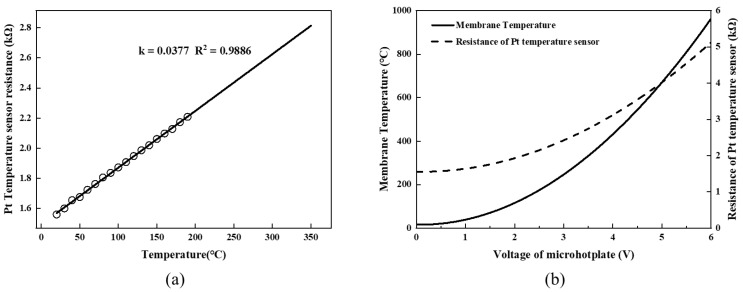
Electric-thermal performance. (**a**) TCR of Pt temperature sensor, (**b**) membrane temperature and resistance of Pt temperature sensor with sweeping voltage.

**Figure 10 micromachines-13-02090-f010:**
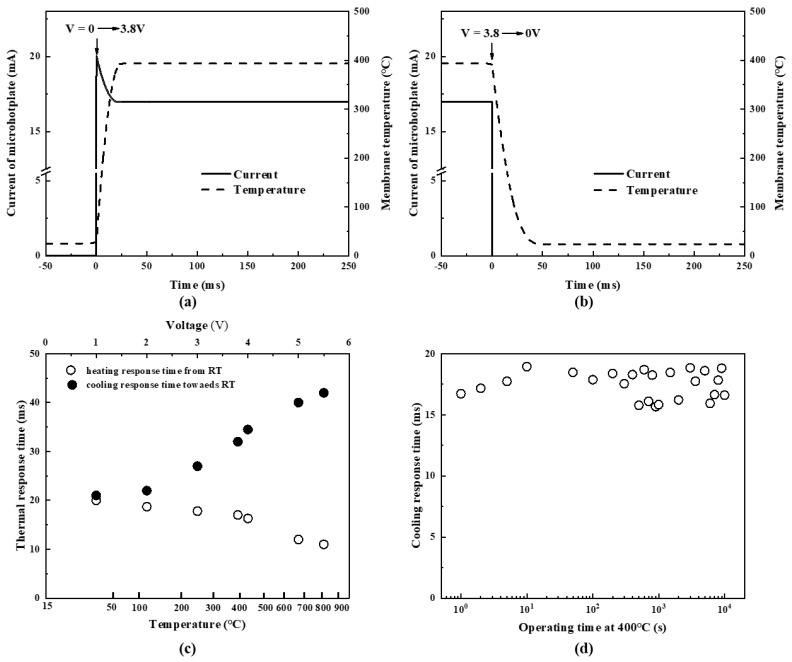
Thermal response characteristic: (**a**) heating from room temperature to 400 °C; (**b**) cooling from 400 °C to room temperature; (**c**) heating and cooling response time with sweeping voltage; (**d**) cooling response time while operating at 400 °C for an hour.

**Table 2 micromachines-13-02090-t002:** Material parameter used in simulation.

Material	Density (Kg/m^3^)	Young’s Modulus (GPa)	Poisson’s Ratio	Thermal Conductivity (W/(m*K))	Thermal-Expansion Coefficient (10^−6^/K)
Si	2329	170	0.28	147	2.6
SiN*_x_*	3100	250	0.23	4.2	2.3
poly-Si	2320	160	0.22	37	2.6

## Data Availability

Not applicable.

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
