# Peer review of "Design and Fabrication of a Novel Poly-Si Microhotplate with Heat Compensation Structure"

_micromachines, 2022, doi:10.3390/mi13122090_

Round 1
Reviewer 2 Report
1. Abstract is not well written, and not in a perfect scientific manner. It is confusing. The authors must provide the findings in a simple way. The utilization areas must be detailed in the introduction and simply at the beginning of the abstract. The abstract should be better at showing the findings in the paper.
2. "rapidly response" should be "rapid response", please change it.
3. "Gas sensors typically operated at high temperatures..." at what please indicate a reference range here with reference?
4. "For gas sensors, the resistive heaters and detection structure generally used the same set of Pt electrode [10,11]."
The references used here in this sentence are not well related to what is said in the sentence.
Please following papers on the matter;
Platinum thin film electrodes for high-temperature chemical sensor applications, Sensors and Actuators B: Chemical, Volume 181, May 2013, Pages 702-714.
Platinum–zirconium composite thin film electrodes for high-temperature micro-chemical sensor applications, Sensors and Actuators B: Chemical, Volume 207, Part A, February 2015, Pages 206-215.
5. What is the targeted working temperature for the heaters designed here?
6. "Therefore, it was essential for 49 enhancing the sensitivity by separating the resistive heater and detection electrodes."
In a typical configuration, those all should not be physically touching each other.
7. "Generally, Pt was chosen to fabricate the heater resistance. "
Please explain why simply? This explanation is important.
8. The introduction is not stating the problem clearly. it is confusing, please answers those questions and organize the introduction.
what is the problem with platinum, gold, or other metals?
what is the requirement for the low-temperature deposition of SiN?
9. what is "a temperature compensated structure was designed and fabricated"? This is a very important and well-defined structure as I can see on Figure 2. Please define this well-structures well in the writing.
10. The introduction, especially on page 2, must be rewritten in a clear and understandable manner.
Reviewer 3 Report
In this work, the authors report the fabrication of a MEMS micro-hot plate that could be heated to high temperatures. They outline the issues with PECVD SiN and argue that LPCVD SiN fares better in various instances. They use Si resistors that are stable during LPCVD SiN growth. Further, they employ a variable resistor design to engineer the distributed heating in the resistors. The premise is interesting and well laid out.
However, I do not recommend the publication of this manuscript due to the following concerns:
11) The ideas introduced in this work are not entirely original, as Hwang et al. (Sensors 2011, 11(3), 2580-2591) reported a similar design with the same functionality and fabrication process. The authors should appropriately cite this work and comment on how their system is improved with respect to Hwang et al.
22) I have several concerns with the simulations. First, the details of the finite element model are not presented. The heat transfer in such micro-heaters has contributions from conduction, convection, and radiation. The parameters for each of these modes must be carefully chosen. Specifically, radiation heat transfer is significant at these temperatures. The authors should briefly describe the parameters that they use for modeling the heat transfer. Interestingly, the temperature rise of the heater, as measured experimentally and as computed using their model agrees quite well with each other. This is not possible if radiation or convection is not included in the model. Also, the model does not include platinum resistor while the fabricated device has a platinum resistor. The presence of this platinum heater affects how much the device heats up at a certain voltage.
33) Further, they discuss the deformation of the membrane in their finite element analysis. However, Pt is not included in their model. Pt is critical because of a mismatch in the thermal expansion coefficients with the underlying materials. The results reported in Fig. 6 are thus misleading unless Pt is included in the modeling. The authors should at least clearly mention the assumptions and approximations in their model.
44) The infrared images reported in Fig. 8 are unclear. First, they should use the same color bar for both figures for a fair comparison. The bright white color in Fig. 8b seems to be due to saturation and not because the whole membrane is at the same temperature. In fact, a significantly large contrast from orange in Fig. 8a to bright white in Fig. 8b is unconvincing, especially because the temperatures differ only by a few tens of degrees. Similar thermographs can be seen in Fig. 6 of Hwang et al., which show reasonable temperatures along with temperature gradients. In the current work, it seems like there are no temperature gradients in both figures which is not possible (even their simulations show gradients).
55) How is the response time defined? Is it the time taken to reach 63% of the final value? If so, then please mention this in the manuscript. Also, do they have any explanation for the difference in cooling and heating time constants? It is particularly interesting to see a different trend for cooling and heating in Fig. 10a.
Other minor suggestions:
11) Overall, the writing can be improved. It is difficult to follow the logic in a few phrases. I note some of them here:
a. first page, ‘rapid response’ instead of ‘rapidly response’
b. At several places, ‘planner’ should be replaced by ‘planar’
c. The term ‘thermal dilation’ is unclear to me. If they mean ‘thermal expansion’, then please replace it.
22) On line 186, the major axis value is smaller than the minor axis value. Please report it appropriately.
33) For the finite element calculations, they seem to have considered only the bulk properties. Thin films are known to exhibit mechanical and thermal properties different from that of the bulk. It is maybe beneficial to comment on this in the manuscript.
44) What is the gap between independent resistors? This may be mentioned in Fig. 4b.
Round 2
Reviewer 3 Report
The authors sufficiently addressed all my comments, and they made significant changes that made the work technically more accurate, reproducible, and readable. I recommend the publication of this manuscript.